# Systemic Oxidative Stress in Severe Early-Onset Fetal Growth Restriction Associates with Concomitant Pre-Eclampsia, Not with Severity of Fetal Growth Restriction

**DOI:** 10.3390/antiox13010046

**Published:** 2023-12-26

**Authors:** Marjon E. Feenstra, Martin F. Bourgonje, Arno R. Bourgonje, Mirthe H. Schoots, Jan-Luuk Hillebrands, Anneke C. Muller Kobold, Jelmer R. Prins, Harry van Goor, Wessel Ganzevoort, Sanne J. Gordijn

**Affiliations:** 1Department of Gynecology and Obstetrics, University Medical Center Groningen, University of Groningen, 9713 GZ Groningen, The Netherlands; m.e.feenstra@umcg.nl (M.E.F.); j.r.prins@umcg.nl (J.R.P.); 2Department of Pathology and Medical Biology, University Medical Center Groningen, University of Groningen, 9713 GZ Groningen, The Netherlands; m.f.bourgonje@umcg.nl (M.F.B.); a.r.bourgonje@umcg.nl (A.R.B.); m.h.schoots@umcg.nl (M.H.S.); j.l.hillebrands@umcg.nl (J.-L.H.); h.van.goor@umcg.nl (H.v.G.); 3Department of Laboratory Medicine, University Medical Center Groningen, University of Groningen, 9713 GZ Groningen, The Netherlands; a.c.muller@umcg.nl; 4Department of Gynecology and Obstetrics, Amsterdam University Medical Centers, Amsterdam Reproduction and Development Research Institute, University of Amsterdam, 1105 AZ Amsterdam, The Netherlands; j.w.ganzevoort@amsteramumc.nl

**Keywords:** oxidative stress, redox status, biomarkers, fetal growth restriction, free thiols, placental insufficiency

## Abstract

Background: Placental insufficiency is an important mechanism underlying early-onset fetal growth restriction (eoFGR). Reduced placental function causes impaired metabolic and gaseous exchange. This unfavorable placental environment is among other processes characterized by increased oxidative stress. Systemic free thiols (FT) are known for their reactive oxygen species scavenging capacity, and higher plasma levels of FT are associated with a better outcome in a multitude of ischemic and inflammatory diseases. We aimed to investigate the relationships between systemic FT levels and maternal and perinatal clinical characteristics and outcomes. Study design: In a post hoc analysis of the Dutch Strider study, a cohort of women with eoFGR, we investigated the association between the maternal redox status (FT) levels at study inclusion, placental biomarkers, and maternal and neonatal outcomes in 108 patients. Results: FT were significantly lower in pregnancies complicated with eoFGR with concurrent maternal hypertensive disorders (pregnancy-induced hypertension; ρ = −0.281 *p* = 0.004, pre-eclampsia; ρ = −0.505 *p* = 0.000). In addition, lower FT levels were significantly associated with higher systolic (ρ = −0.348 *p* = 0.001) and diastolic blood pressure (ρ = −0.266 *p* = 0.014), but not with the severity of eoFGR. FT levels were inversely associated with sFlt (ρ = −0.366, *p* < 0.001). A strong relation between systemic FT levels and PlGF levels was observed in women with pre-eclampsia at delivery (ρ = 0.452, *p* = 0.002), which was not found in women without hypertensive disorders (ρ = 0.008, *p* = 0.958). Conclusions: In women with pregnancies complicated with eoFGR, FT levels reflect the severity of maternal disease related to the underlying placental insufficiency rather than the severity of the placental dysfunction as reflected in eoFGR or perinatal outcomes.

## 1. Introduction

Early-onset fetal growth restriction (eoFGR, <32 weeks of gestation) is a rare condition, affecting 0.3% of all pregnancies [1]. The disorder is associated with (iatrogenic) preterm birth and a range of adverse perinatal and long-term outcomes, including stillbirth, neonatal morbidity and mortality, neurodevelopmental disorders, and a higher risk of metabolic syndrome in adult life [2,3,4,5]. eoFGR often coincides with the development of maternal hypertensive disease during pregnancy, including severe early-onset pre-eclampsia, which aligns with shared underlying placental pathology. In normal placental development, first trimester remodeling of maternal spiral arteries results in a low-resistance, high-volume flow in the intervillous space for optimal nutrient and oxygen supply to the fetus [6,7,8]. Inadequate/superficial remodeling results in maternal vascular malperfusion (MVM) in the placenta [7,9,10,11]. The inadequate remodeling results in early intervillous perfusion under relative high pressure, which leads to ischemia-reoxygenation damage and altered villous vascular development with impaired metabolic and gaseous exchange between mother and fetus [12,13]. Fetal malnutrition and hypoxia result in small fetal size, abnormal vascular resistance, and compensatory mechanisms to preserve the function of the most vital organs, as seen in Doppler flow profiles and abnormal fetal heart rate patterns [14,15]. The unfavorable placental condition results in altered cellular homeostasis with increased apoptosis and senescence [16,17]. The associated increase in placental waste products in the maternal circulation induces a maternal endothelial response that can be clinically recognized as pre-eclampsia (Figure 1) [18].

Oxidative stress is evidenced by increased levels of reactive oxygen species (ROS) [19]. Systemic free thiols (FT) are known for their ROS scavenging capacity and are an important component of the in vivo antioxidant buffer capacity, thereby playing a protective role against oxidative stress [20,21,22]. Oxidative stress can be systemically detected by the depletion of plasma FT, since these are readily oxidized by ROS. Reduced levels of FT are frequently observed in human diseases that are typically associated with oxidative stress (e.g., chronic kidney disease, chronic heart failure, diabetes, cancer, systemic sclerosis, and inflammatory bowel disease), while higher concentrations of FT are associated with a more favorable outcome in healthy individuals, patients with chronic heart failure, and renal transplant recipients [23,24,25]. Furthermore, systemic levels of FT show an inverse relationship with hypertension in the general population [26]. PlGF is used as a marker for syncytiotrophoblast remodeling and is known as a biomarker for pre-eclampsia. Inadequate remodeling of the spiral arteries could lead to hypoxia-reoxygenation damage and an environment that increases placental oxidative stress [27,28]. sFlt is known to increase with endothelial dysfunction in the placenta, which could subsequently increase oxidative stress. Both sFlt and PlGF are implemented biomarkers of FGR and pre-eclampsia [27,29].

We hypothesized that plasma FT, as a marker of systemic oxidative stress, has merit as a read-out of the chance and severity of developing adverse outcomes in eoFGR and the interaction with (development of) maternal hypertensive disorders. This has been studied for pregnancies complicated by pre-eclampsia but not for eoFGR [21,30].

Therefore, we aimed to investigate the interrelationships between systemic FT levels and maternal and perinatal clinical characteristics and outcomes, including the development of maternal hypertensive disease and the extent of eoFGR. 

## 2. Materials and Methods

This study was a post hoc analysis of the Dutch STRIDER trial, including all participants with available maternal blood samples at study inclusion prior to the initiation of study medication [31].

### 2.1. Original Study Design

The Dutch Sildenafil TheRapy In Dismal prognosis Early-onset fetal growth Restriction (STRIDER) study was a placebo-controlled randomized clinical trial in 11 centers in the Netherlands. The study evaluated the use of sildenafil citrate as a therapy for eoFGR. Sildenafil citrate is a phosphodiesterase type 5 inhibitor that blocks the inactivation of cyclic guanosine monophosphate (cGMP) in vascular smooth muscle cells, increasing NO-mediated vasodilation [31,32,33]. The study methods have been described in detail elsewhere [31]. In summary, pregnant women between 20 weeks and 0 days and 29 weeks and 6 days of gestation with eoFGR were randomized to receive sildenafil citrate 25 mg three times daily or a placebo. eoFGR was defined as an abdominal circumference below the 3rd percentile or the estimated fetal weight (EFW) below the 5th percentile, combined with either unilateral or bilateral notching of the uterine artery, the Pulsatility Index (PI) of the umbilical artery above the 95th percentile, the PI of the middle cerebral artery below the 5th percentile, or the presence of a maternal hypertensive disorder until 27 weeks and 6 days of gestation. From a gestational age of 28 weeks on, eoFGR was defined as an EFW of less than 700 g, combined with, as previously stated, Doppler anomalies and/or the presence of maternal hypertensive disorder. The EFW was calculated by using the Hadlock equation.

The Dutch STRIDER was halted prior to completion (after inclusion of 216 cases of 354 intended, Figure 2) based on an interim analysis revealing more pulmonary hypertension in the infants born in the sildenafil citrate group in the context of futility (no reasonable potential for a net positive effect on perinatal outcomes). Sildenafil citrate compared with placebo did not reduce the risk of perinatal mortality or major neonatal morbidity, in line with the observations in simultaneous STRIDER trials in other countries. For the current post hoc analysis, 108 women with eoFGR were included, from whom maternal blood samples at inclusion were available for analysis. We used all relevant clinical characteristics. For the EFW and abdominal circumference, absolute values have been used because most EFW and abdominal circumference for gestational age (EFW and abdominal percentile) values are below the 1st percentile. The birth weight ratio was calculated as the ratio of the observed birth weight with the median birth weight for the gestational age at birth.

### 2.2. Serum Sample Collection

In a subset of centers with a biomaterial storage facility, maternal serum samples (n = 108) of all participants were collected at study inclusion before the start of the intervention and stored at −80 °C until the current secondary analysis of FT. All samples available were used for this study. FT levels have been presented as continuous values. 

### 2.3. Laboratory Measurements

FT concentrations were measured as described previously. (34) After thawing, samples were four-fold diluted using 0.1 M Tris buffer (pH 8.2). Background absorption was determined at 412 nm using the Varioskan microplate reader (Thermo-Scientific, Breda, The Netherlands), alongside a reference measurement at 630 nm. Next, 20 μL of 1.9 mM 5,5′-dithio-bis-2-nitrobenzoic acid (DTNB, Ellman’s reagent, CAS no. 69-78-3, Sigma-Aldrich Corporation, St. Louis, MO, USA) was added to the samples in 0.1 M phosphate buffer. After 20 min of incubation at room temperature, the sample absorbance was measured again. Final concentrations of plasma FT were established by parallel measurement of an L-cysteine calibration curve (CAS no. 51-90-4, Fluka Biochemika, Buchs, Switzerland) in a concentration range of 15.6–1000 μM in 0.1 M Tris/10 mM EDTA (pH 8.2). Intra- and inter-assay coefficients of variation were all below 10%.

Placental growth factor (PlGF) and soluble fms-like tyrosine kinase (sFlt) levels were measured on the Kryptor immunoassay (Thermo Fisher Scientific) and compared with the fifth percentile of a reference population (i.e., 106.54 pg/mL) [34].

### 2.4. Statistical Analysis

Baseline characteristics of the study population were presented as mean ± standard deviation (SD), median (interquartile range, IQR), or proportions n with percentages (%). Normality assessment was performed by visual inspection of histograms and normal probability (Q–Q) plots. Comparisons between groups for continuous variables were performed using independent sample *t*-tests or Mann–Whitney U tests (depending on normality), while for nominal variables, chi-square tests or Fisher’s exact tests were performed, as appropriate. Correlations between continuous variables were calculated using Spearman’s rank correlation coefficients (ρ).

Data analysis was performed using SPSS (Statistics 28.0 software package, (SPSS Inc., Chicago, IL, USA) and the Python programming language (v.3.8.8, Python Software Foundation, https://www.python.org, accessed on 5 May 2023) using the pandas (v.1.2.3) and numpy (v.1.20.0) packages. Data visualization was performed using the seaborn (v.0.11.1) and matplotlib (v.3.4.1) packages. Two-tailed *p*-values < 0.05 were considered statistically significant. 

### 2.5. Ethical Considerations

Ethical approval for the STRIDER study was given on 22 July 2014 (2014-131). Trial registration number: ClinicalTrials.gov identifier: NCT02277132 (registered 29 September 2014).

## 3. Results

### 3.1. Cohort Characteristics 

The clinical characteristics of this cohort are presented in Table 1. Blood samples were available from 108 of the 216 women (50%) included in the original trial. Baseline characteristics of the women in this cohort and the original Dutch STRIDER RCT cohort are given in Appendix A [31].

### 3.2. Oxidative Stress: Plasma Free Thiols (FT) as a Biomarker

At inclusion, lower FT levels were associated with pre-eclampsia (ρ = −0.505 *p* < 0.001) but not with gestational hypertension (ρ = −0.085 *p* = 0.383), whereas the lower FT levels at inclusion were associated with both maternal hypertensive disorders and the development of pre-eclampsia (ρ = −0.313 *p* = 0.002, Figure 3B) and gestational hypertension (ρ = −0.281 *p* = 0.004, Figure 3A). Overall, lower FT levels were associated with higher blood pressure [both systolic blood pressure (ρ = −0.348 *p* = 0.001, Figure 3C) and diastolic blood pressure (ρ = −0.266 *p* = 0.012, Figure 3D)] and with increased levels of proteinuria (ρ = −0.660 *p* = 0.004, Appendix A). A later gestational age was associated with lower FT levels (ρ = −0.343 *p* < 0.001) (Appendix A), as were higher EFW in grams (ρ = −0.450 *p* < 0.001), and larger fetal abdominal circumference in mm (ρ = −0.404 *p* < 0.001). 

FT levels were not associated with abnormal uterine artery flow profiles, fetal Doppler flow patterns, or birthweight ratio (ρ = −0.149, *p* = 0.131) (Figure 4D–E, Appendix A). 

FT levels did not differ between different modes of delivery, but lower FT levels were significantly associated with induction of labor and prelabor caesarean section for maternal indication (e.g., hypertension, pre-eclampsia, and HELLP, ρ = 0.270, *p* = 0.007, Appendix A). FT levels did not differ between live births and stillbirths (ρ = −0.051, *p* = 0.603) or neonatal deaths. FT levels were not associated with birthweight (ρ = 0.056, *p* = 0.473), neonatal length (ρ = −0.103, *p* = 0.473), head circumference (ρ = 0.159, *p* = 0.301) at birth, placental weight (ρ = 0.090, *p* = 0.989), or any individual type of neonatal morbidity (Figure 5A–G). Maternal demographics, BMI, and fetal sex were also not associated with FT levels (Appendix A).

We observed significant associations between systemic FT levels and the biomarkers sFlt and PlGF (Figure 6). sFlt levels were inversely related to systemic FT levels in all women in this cohort with and without pre-eclampsia at the time of delivery (ρ = −0.366, *p* < 0.001, Figure 6A). Similarly, we observed a strong relationship between systemic FT levels and PlGF levels in women with pre-eclampsia at delivery (ρ = 0.452, *p* = 0.002), which was not found in women without hypertensive disorders (ρ = 0.008, *p* = 0.958, Figure 6B).

## 4. Discussion

In women with pregnancies complicated by eoFGR, FT levels did not differentiate between the severity of eoFGR or the perinatal outcomes. FT levels differentiated between those with and without pre-eclampsia, both known at the time of diagnosis or developing later, the development of maternal hypertensive disorders, and the subsequent induction of labor due to maternal hypertensive disorders. FT levels are strongly related to PlGF levels and inversely related to sFlt levels. We conclude from these results that systemic oxidative stress in pregnancies complicated by eoFGR with concurrent (development of) maternal hypertensive disorders is likely to be a reflection of the maternal vascular status and (development- and severity-related) hypertensive disease related to the development of placental disease resulting in eoFGR rather than the severity of the placental dysfunction in response to the underlying placental disorder.

eoFGR is known for the coexistence of (and development of) maternal hypertensive disorders with the commonality in the predominant underlying pathophysiological process of impaired spiral artery remodeling in early gestation, resulting in placental MVM, ischemia-reperfusion damage, and increased placental senescence [35]. Among women with early-onset pre-eclampsia, FGR is seen in more than 90%, and among women with eoFGR, hypertension is seen in approximately 50% [3]. These sequelae were also observed in our study, and therefore, if women are diagnosed with eoFGR, they can be considered to be at very high risk for the development of hypertensive disease if they are not already present at the time of diagnosis.

For this study, FT levels in maternal serum samples at the study inclusion of the Dutch STRIDER cohort were used. Because samples were taken before the start of the admission of sildenafil citrate or placebo and because the study arm allocation had no effect on the relevant outcomes, we assume that treatment allocation had no effect on our analyses. Unfortunately, no maternal blood samples were taken after inclusion. 

The selection of the subpopulation for the post hoc analysis was based on the institutional availability of maternal blood samples and storage facilities and may therefore reflect some population or practice variation that is likely to have a limited influence on the outcomes of this analysis. 

Our findings that systemic oxidative stress is mainly related to maternal hypertension and not a reflection of pathophysiological processes in the placenta are consistent with previous observations of the relationship between FT levels and the development of hypertension in the general population [23]. Also, a previous pilot study conducted in our center showed similar results [34].

We conclude that FT can potentially differentiate pregnancies complicated by eoFGR with or without the development of maternal hypertensive disease. The strong associations between maternal hypertensive disorders and systemic oxidative stress correspond well to what is known about the pathophysiology of hypertension [36,37]. More known placental biomarkers (PLGF and sFlt-1) show a strong association with FT. sFlt levels were inversely related to systemic FT levels in all women in this cohort with and without pre-eclampsia at the time of delivery. Similarly, we observed a strong relationship between systemic FT levels and PlGF levels in women with pre-eclampsia at delivery, which was not found in women without hypertensive disorders. Which corresponds with current literature [29]. Therefore, FT could possibly be used as an addition to a biochemical profile in pregnancy with the more known placental biomarkers. 

More research needs to be performed to investigate if FT could be used in eoFGR as a predictor for the development of pre-eclampsia and hypertensive disease or in pregnancies with women at risk for eoFGR with and without pre-eclampsia. And more research is needed if sequential measurements could be used for a prediction of delivery indication on maternal grounds. This cohort lacked healthy controls, and there is no current literature about the predictive value of FT for pre-eclampsia or hypertension in pregnancy. Prospective cohorts of various risk profiles linking FT levels to complications of pregnancy would be valuable future research. Tan et al., (2018) describe a screening program for pre-eclampsia (SPREE). They state that screening approaches for pre-eclampsia in the low-risk/high risk/very high population currently recommended are poor and could be improved by adding biomarkers such as PlGF [27,29]. In our current high-risk cohort, FT levels have similar predictive features. Therefore, we suggest that more research into using FT levels as a biomarker in screening panels could be useful in assessing risk profiles for maternal hypertensive disorders to improve screening and subsequent pregnancy management. The current study provides a basis for the potential use of FT in screening panels but is not sufficient for its implementation in clinical practice.

The strength of this study is that it was a prospectively collected cohort of eoFGR from the patients of the Dutch STRIDER study, for whom blood samples were available. Given that this disorder is rare, the size of the cohort was substantial. Also, the number analyzed was higher than in previously published studies, and the prospective data quality was high [34].

Some limitations need to be mentioned. The total number of samples (n = 108) was limited to the cases in which blood samples were collected in centers with the capability to store the maternal blood samples. The resulting potential for selection bias is probably limited because selection was not patient-specific. Another limitation of this study is that we did not have a control group without FGR matched for gestational age at delivery. Also, we have no information from subsequent samples after study inclusion and near delivery to analyze the effect of sildenafil and changes in systemic oxidative stress. Also, we could not relate FT levels just before birth to neonatal outcomes because we did not take blood samples at delivery or umbilical cord blood to analyze systemic oxidative stress in the neonate. And last, we measured only one of the many markers of oxidative stress, thereby potentially missing some of the nuance of the changes in the redox constellation, and due to limited serum samples, FT levels could not be corrected for albumin. 

## 5. Conclusions

Increased systemic oxidative stress, as represented in this current study by low FT levels in maternal serum samples in eoFGR pregnancies, is associated with the maternal syndrome but not with the severity of the fetal consequences of the placental dysfunction. These results help to understand the pathophysiology of eoFGR with and without maternal hypertensive disorders.

## Figures and Tables

**Figure 1 antioxidants-13-00046-f001:**
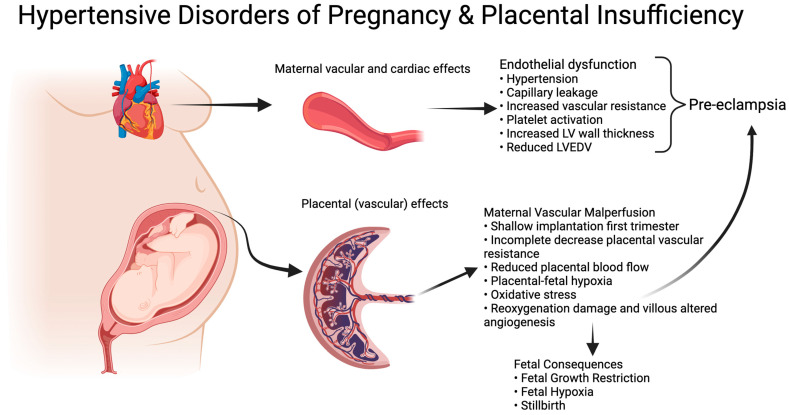
Hypertensive Disorders of Pregnancy and Placental insufficiency. Created with BioRender.com.

**Figure 2 antioxidants-13-00046-f002:**
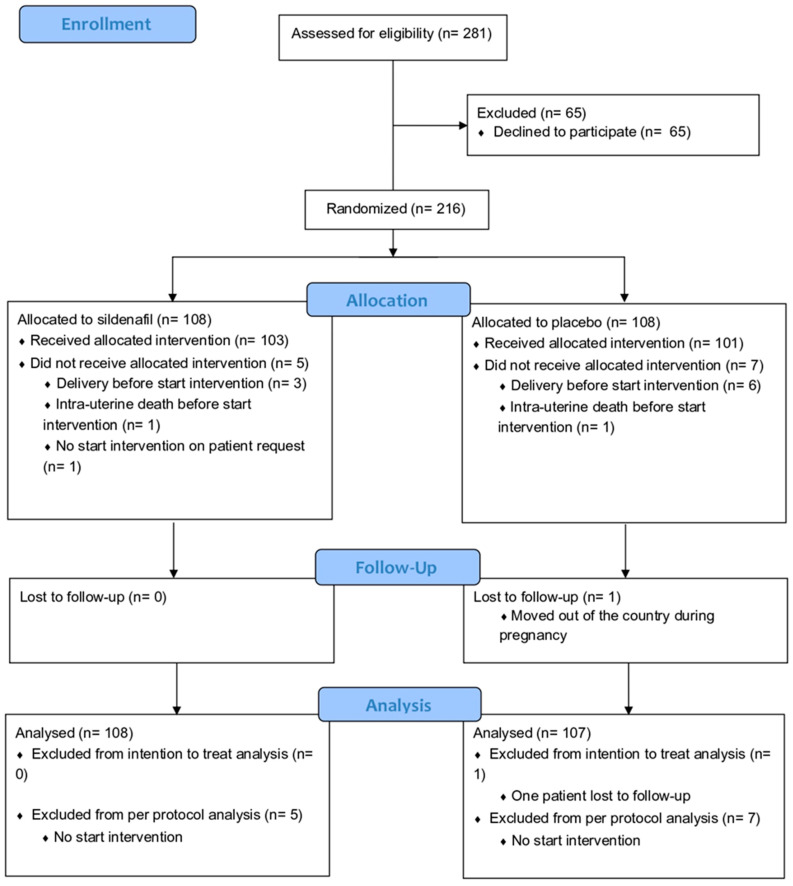
Consort flow diagram of original Dutch STRIDER randomized controlled trial [2].

**Figure 3 antioxidants-13-00046-f003:**
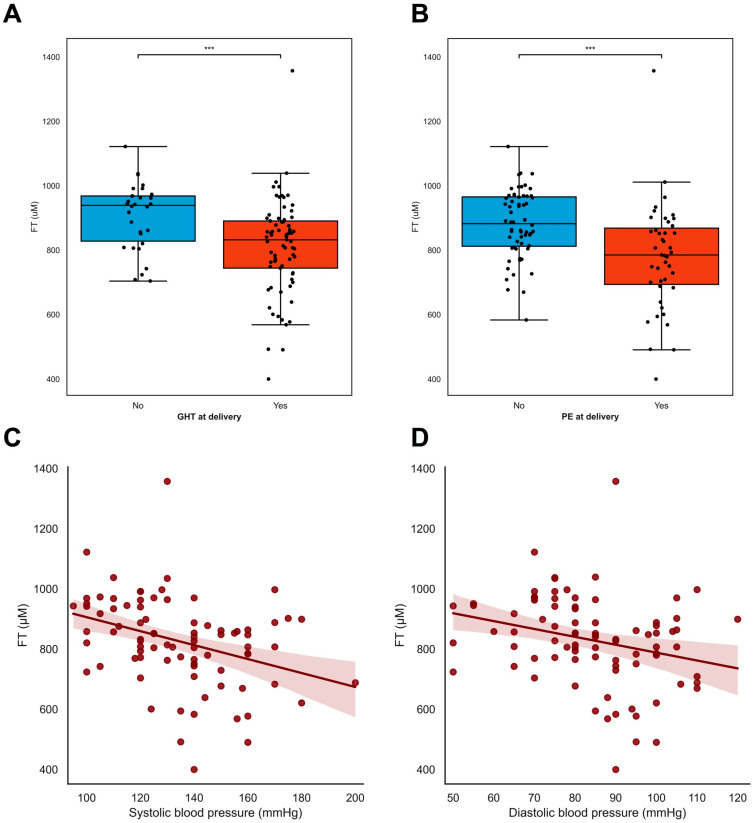
FT levels in relation to maternal hypertensive disorders. (**A**–**D**) FT levels were associated with maternal hypertensive disorders in pregnancy. (**A**,**B**) Pregnant women with either hypertension or pre-eclampsia at inclusion (**A**) and time of delivery (**B**) had lower levels of free thiols compared to women without (development of) hypertension or PE in pregnancies. Black dots resemble individual cases. Boxplots were drawn according to the Tukey method, with inner boundaries defined as the 25th/75th percentiles ± 1.5 IQR. (**C**,**D**) Systolic and diastolic blood pressure at baseline were inversely associated with plasma free thiol levels in pregnancy. *** *p* < 0.001. Abbreviations: FGR; fetal growth restriction; PE; pre-eclampsia; FT; total free thiols; PIH; pregnancy-induced hypertension.

**Figure 4 antioxidants-13-00046-f004:**
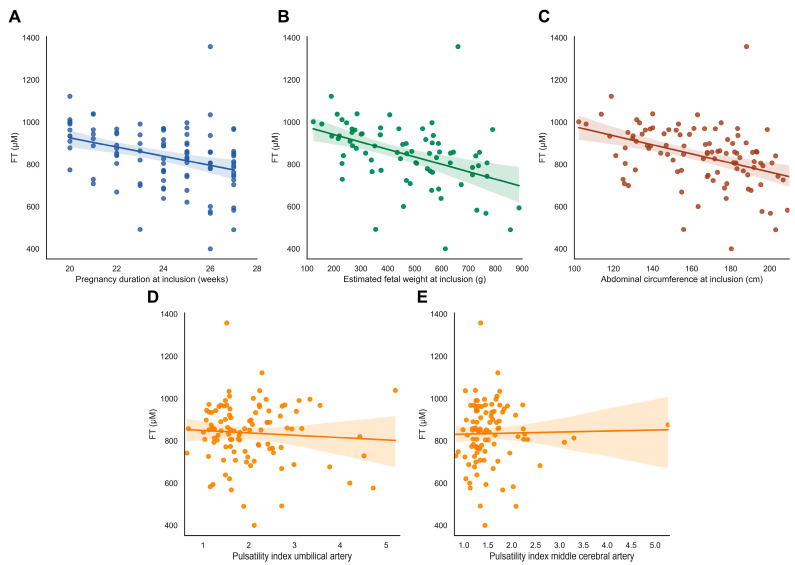
FT levels in relation to fetal outcomes. (**A**–**E**) FT levels in relation to fetal characteristics. (**A**) Gestational age at inclusion in the Dutch STRIDER study. (**B**,**C**) Estimated fetal weight (EFW) in grams and fetal abdominal circumference (AC) in mm at inclusion. (**D**,**E**) Pulsatility index of the umbilical artery (UA) and middle cerebral artery (MCA) at inclusion. Abbreviations: FGR, fetal growth restriction; PE, pre-eclampsia; FT, total free thiols.

**Figure 5 antioxidants-13-00046-f005:**
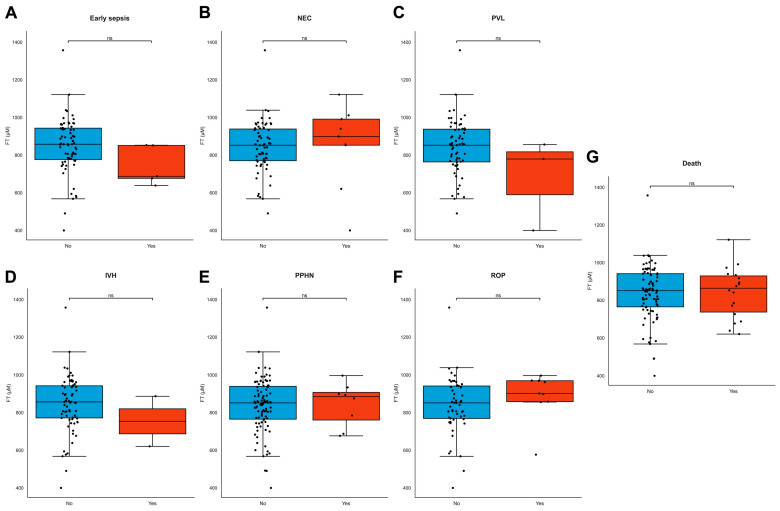
(**A**–**G**) FT levels in neonatal adverse outcomes. (**A**–**G**) FT levels did not significantly differ across common neonatal pathologies. Boxplots demonstrating distributions of free thiol levels in the presence and absence of common neonatal pathologies, including early sepsis (**A**), necrotizing enterocolitis (NEC) (**B**), periventricular leukomalacia (PVL) (**C**), intraventricular hemorrhage (IVH) (**D**), persistent pulmonary hypertension of the newborn (PPHN) (**E**), retinopathy of prematurity (ROP) (**F**), and neonatal death (**G**). Boxplots were drawn according to the Tukey method, with inner fences defined as the 25th/75th percentiles ± 1.5 IQR. Black dots represent individual cases. Significances were calculated according to Mann–Whitney U-tests with post hoc Bonferroni correction for multiple comparisons. Abbreviations: FT, free thiols; ns, non-significant.

**Figure 6 antioxidants-13-00046-f006:**
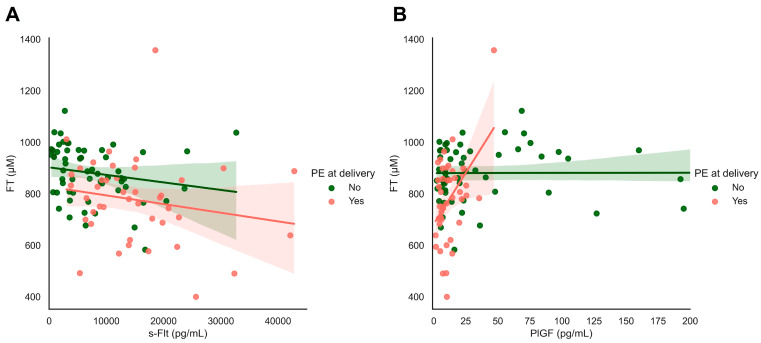
s-Flt and PIGF in relation to FT and maternal outcomes: focusing on maternal hypertensive disorders. (**A**,**B**) Plasma-free thiols (FT) were associated with sFlt and PlGF levels as classical biomarkers of FGR/PE. Both s-Flt and PlGF were nominally significantly associated with plasma FT levels in the full study cohort.

**Table 1 antioxidants-13-00046-t001:** Clinical characteristics. Data are presented in proportions n with corresponding percentages (%) or median with IQR unless stated otherwise in the table. Abbreviations: BMI, body mass index (calculated as weight in kilograms divided by height in meters squared). The estimated fetal weight (EFW) was calculated by using the Hadlock equation. Pre-eclampsia was defined as gestational hypertension accompanied by one or more of the following new-onset conditions at or after 20 weeks’ gestation: 1. Proteinuria 2. Other maternal organ dysfunctions 3. Uteroplacental dysfunction [35]. # Study medication was given after the collection of maternal blood.

Baseline	(n = 108)
Maternal	
Age (years)	32 (±5)
BMI (kg/m^2^)	24.5 (±6.1)
Ethnicity	
- Caucasian - African - Asian - Other	80 (79.2%) 11 (10.9%) 3 (3.0%) 14 (13%)
Current smoker	8 (7.6%)
Low-dose aspirin usage	10 (9.5%)
Gestational hypertension	23 (21.9%)
Pre-eclampsia	21 (20%)
Systolic blood pressure (mm Hg)	134 (±41)
Diastolic blood pressure (mm Hg)	84 (±24)
Sildenafil #	52 (50.5%)
FetaI	
Gestational age (weeks + days, IQR)	24 + 3 (23 + 3 to 25 + 3)
Estimated fetal weight (grams)	517.2 (±338.3)
Abdominal circumference (mm)	171.3 (±42.3)
Umbilical artery *p* > p95	61 (59.2%)
Middle cerebral artery pulsatility index: <p5	58 (59.8%)
Birth outcomes	
Gestational hypertension	73 (67.6%)
Gestational hypertension developed after inclusion	40 (37.0%)
Pre-eclampsia	43 (39.8%)
Pre-eclampsia developed after inclusion	22 (20.4%)
Onset of labor	
Spontaneous	9 (8.6%, 5 stillbirths)
Induction of labor	35 (32.4%, 24 stillbirths)
Prelabour cesarean section	61 (58.1%, 0 stillbirths)
Indication for induced delivery (n = 96)	
Maternal	23 (24%)
Fetal indication	73 (76%)
Mode of delivery	
Spontaneous vaginal delivery	40 (38.1%, 29 stillbirths)
Instrumental vaginal delivery	1 (1.0%, 0 stillbirths)
Caesarean section	64 (61%, 0 stillbirths)
Neonatal outcomes	
Gestational age at delivery (weeks + days)	28 + 3 (21 to 38)
Birthweight (g)	1170 (±1215)
Birthweight ratio	0.5583 (±0.136)
Stillbirth	29 (27.6%)
Male sex	58 (53.7%)
Cord pH is arterial	7.3 (±0.10)
Cord pH: venous	7.4 (±0.13)
Birthweight below p10	86 (96.6%)
Birthweight below p3	77 (86.5%)
Persistent pulmonary hypertension	8/79 (10.1%)
Necrotizing enterocolitis	9/79 (11.3%)
Early-onset sepsis	5/79 (6.3%)
Late-onset sepsis	20/79 (25.3%)
Periventricular leukomalacia grade ≥ 3	0/79 (0%)
Intraventricular hemorrhage grade ≥ 3	2/79 (2.5%)
Neonatal death	18/79 (22.7%)

## Data Availability

Data are available upon request. Correspondence and requests for materials should be addressed to S.J.G. or W.G.

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
