# Peer review of "Systemic Oxidative Stress in Severe Early-Onset Fetal Growth Restriction Associates with Concomitant Pre-Eclampsia, Not with Severity of Fetal Growth Restriction"

_antioxidants, 2023, doi:10.3390/antiox13010046_

Round 1

Reviewer 1 Report

Comments and Suggestions for Authors

1. This is an interesting post hoc analysis.

2. The results and possible use will be limited to pregnancy hypertension related researchers.

3. The separation of the maternal vascular effects vs placental dysfunction is an important distinction for the average reader so the different cellular source locations and responses could be highlighted in a figure or diagram as most OB providers understand or blame the placenta for all this hypertensive and fetal morbidity but you indicate the separation.

4. You have summarized the strengths and limitations very well.

5. The discussion re adding FT to a 'screening' or 'diagnostic' panel  must be expanded as most providers have difficulty understanding the patient counselling issues related to screening from diagnosis.

Comments on the Quality of English Language

No large concerns.

Author Response

We thank the reviewer for his/her comments, our response is in the attached file. 

Reviewer 2 Report

Comments and Suggestions for Authors

p. 2. l. 71-72. Briefly describe the relationship between oxidative stress and PlGF / sFLT.

Was the free thiol level measured from blood or serum? In the methodological part, it is about blood, in the referenced publications there is thiol determination from serum.

Was the free thiol level corrected for albumin concentration?

The manuscript also mentions that there is no control group This strongly limits the usability of the results (this statement is omitted from systematic reviews). Measurement of free thiol is low-cost, and clinical data can usually be obtained at minimal cost. The high-cost PlGF and sFLT measurement could be corrected with a low number of samples (15-20). It would help other research groups to make comparisons if at least ethnicity was described. Caucasian, African, and Asian ethnic groups can give very different results.

The meanings of *, **, *** are missing in figure captions!

Correct consistent errors! for example: data collection, data quality was high.[35]  data collection, data quality was high [35].

Author Response

We thank the reviewer for his/her comments. 

2. l. 71-72. Briefly describe the relationship between oxidative stress and PlGF / sFLT.

We appreciate this suggestion, and we added “PlGF is used a as a marker for syncytiotrophoblast remodeling, and is known as a biomarker for pre-eclampsia. Inadequate remodeling of the spiral arteries could lead to a hypoxia-reoxygenation damage and environment, and increase placental oxidative stress.[6,14] sFlt is known to increase with endothelial dysfunction in the placenta, which could subsequently increase to an increase in oxidative stress. Both sFlt and PlGF are implemented biomarkers of FGR and pre-eclampsia.[26]”

Was the free thiol level measured from blood or serum? In the methodological part, it is about blood, in the referenced publications there is thiol determination from serum.

We thankfully corrected maternal blood samples to maternal serum samples.

Was the free thiol level corrected for albumin concentration?

We agree with the reviewer that it would have been ideal to be able to correct for albumin concentration. Unfortunately, no albumin data was available in this cohort. In previous studies from our research group we adjusted for albumin. This adjustment was performed since albumin is the predominant blood protein and therefore it could determine the total quantity of potentially detectable FT groups.1 We do agree with the reviewer that this is valid point, since in certain patient groups this could be responsible for potential differences in systemic redox status (especially those in which albumin perturbances are likely to occur – e.g. sepsis with hypoalbuminemia). However, the patient group within the context of this study consists of a group of women whom were generally healthy before pregnancy and were assumed to have albumin concentrations within the physiological range. Thus, we expect that the unexplained variability due to lack of adjustment for albumin will be relatively minor. Unfortunately, we cannot test for this in our study. It will be an important point for future research. We added in the strengths and limitations “and due to limited serum samples FT levels could not be corrected for albumin.” This is relevant to consider since albumin is the predominant protein in human blood.

  1. Cortese-Krott MM, Koning A, Kuhnle GGC, Nagy P, Bianco CL, Pasch A, Wink DA, Fukuto JM, Jackson AA, van Goor H, Olson KR, Feelisch M. The Reactive Species Interactome: Evolutionary Emergence, Biological Significance, and Opportunities for Redox Metabolomics and Personalized Medicine. Antioxid Redox Signal. 2017 Oct 1;27(10):684-712. doi: 10.1089/ars.2017.7083. Epub 2017 Jun 6. PMID: 28398072; PMCID: PMC5576088.

The manuscript also mentions that there is no control group This strongly limits the usability of the results (this statement is omitted from systematic reviews). Measurement of free thiol is low-cost, and clinical data can usually be obtained at minimal cost. The high-cost PlGF and sFLT measurement could be corrected with a low number of samples (15-20).

We agree with the reviewer, as also stated in our limitations, that a control group would be desirable in most cases. In this study, however, the cost of analysis was not the major problem, more so the availability of a cohort of healthy pregnant women with uncomplicated pregnancies to use as a control cohort. A healthy control group matched to gestational age and demographics is difficult for this group with eoFGR, whereas prematurity is very common. A previous pilot study from our center (Schoots et al. 20211) was a case-control analysis. This study reported three different groups. A healthy control group and two groups with FGR with or without PE. They observed no significant differences between healthy pregnancies and FGR without PE. Those two groups had similar FT level profiles. Herein, the group with FGR combined with PE showed a significantly different FT level profile. This observation is similar to our results in a larger secondary analysis. The aim of this study was to investigate neonatal and fetal outcomes in case of severe eoFGR with and without development of HT disease in pregnancy. For that hypothesis, a control group was not necessary to answer our research question. We do agree with the reviewer that a control group of healthy pregnancies, or pregnancies without FGR could help to comparatively assess systemic oxidative stress in pregnancies complicated and uncomplicated. Therefore, we stated in our limitations that, depending on the specific context, the inclusion of a healthy control group could be useful.  

  1. Schoots MH, Bourgonje MF, Bourgonje AR, Prins JR, van Hoorn EGM, Abdulle AE, Muller Kobold AC, van der Heide M, Hillebrands JL, van Goor H, Gordijn SJ. Oxidative stress biomarkers in fetal growth restriction with and without preeclampsia. Placenta. 2021 Nov;115:87-96. doi: 10.1016/j.placenta.2021.09.013. Epub 2021 Sep 20. PMID: 34583270.

It would help other research groups to make comparisons if at least ethnicity was described. Caucasian, African, and Asian ethnic groups can give very different results.

We thank the reviewer for his/her suggestion. We added the percentage of each ethnicity. Because the population was predominantly of Caucasian descent we did not perform statistical analysis on ethnicity.

The meanings of *, **, *** are missing in figure captions! Correct consistent errors! for example: data collection, data quality was high.[35]  data collection, data quality was high [35]

We thank the reviewer for his/her attentiveness. We corrected the above and checked for other errors.

Reviewer 3 Report

Comments and Suggestions for Authors

Dear Authors,

In the statistical section some parts became confused:   * Advice: If one factor level gives a normality with Shapiro-Wilk’s test (or even with Kolmogorov-Smirnow test) then Pearson r are necessary, but if another factor level is not following the normality then Spearman’s rho is necessary.  

* The authors specified: Correlations between continuous variables were calculated using Spearman's rank correlation coefficients. From the above it is understood that all the continuous variables had not following the normality. The authors must clarify this aspect, because in Table 1 many variables are presented as mean with SD, so they have a normal distribution and their correlations with other normally distributed variables must be done with the Pearson test.

* Scheme 1 should also be checked.
  - in Figure 1, the authors must add to the existing legend, other comments for A-D to be easily understood by all readers:  what do black or red symbols represent ??, the horizontal line at the box level represents mean or median ??, whiskers plot means Min to Max ??, the meaning of *** ?? * See the legend of Figure 3 at the end, where the authors have provided details about the graphs.  

- Also for Figure's 2 and 4, some details about what can be seen in the graphs, not just their title
In the Conclusions section: 

* I suggest that it needs to be improved because it does not present any conclusion of the present study, related to systemic free thiols.  * It is too general in the sentence: Increased systemic oxidative stress in eoFGR pregnancies is associated with the maternal syndrome...; there are many more biomarkers that can be investigated in the context of systemic oxidative stress.

Author Response

We thank the reviewer for his her comments. 

In the statistical section some parts became confused:   * Advice: If one factor level gives a normality with Shapiro-Wilk’s test (or even with Kolmogorov-Smirnow test) then Pearson r are necessary, but if another factor level is not following the normality then Spearman’s rho is necessary.  * The authors specified: Correlations between continuous variables were calculated using Spearman's rank correlation coefficients. From the above it is understood that all the continuous variables had not following the normality. The authors must clarify this aspect, because in Table 1 many variables are presented as mean with SD, so they have a normal distribution and their correlations with other normally distributed variables must be done with the Pearson test.

We thank the reviewer for his/her remarks. For our analysis, we decided to analyze our data as consistent and fairly conservative as possible. We agree with the reviewer that the Pearson correlation test could be used when normality can be assumed while the Spearman’s rho correlation test could be used in cases normality cannot be assumed. Furthermore, the Spearman test has a disadvantage when using large number of data points, albeit in our dataset we had only limited numbers of cases. In our data set also we also had variables with ordinal values. An advantage of the Spearman rank correlation coefficient is that the X and Y values can be continuous or ordinal, and that approximate normal distributions for X and Y are not required. Therefore, we decided to use one statistical test for correlations throughout our analyses.[1]  In addition, the Spearman’s rho correlation test is more robust to possible outliers in our data set.[2] For future purposes, it would also be easier in this way to compare these correlations to other cohort studies.

  1. Bishara AJ, Hittner JB. Testing the significance of a correlation with nonnormal data: comparison of Pearson, Spearman, transformation, and resampling approaches. Psychol Methods. 2012 Sep;17(3):399-417. doi: 10.1037/a0028087. Epub 2012 May 7. PMID: 22563845.
  2. Bishara AJ, Hittner JB. Reducing Bias and Error in the Correlation Coefficient Due to Nonnormality. Educ Psychol Meas. 2015 Oct;75(5):785-804. doi: 10.1177/0013164414557639. Epub 2014 Nov 11. PMID: 29795841; PMCID: PMC5965513.

* Scheme 1 should also be checked.
  - in Figure 1, the authors must add to the existing legend, other comments for A-D to be easily understood by all readers:  what do black or red symbols represent ??, the horizontal line at the box level represents mean or median ??, whiskers plot means Min to Max ??, the meaning of *** ?? * See the legend of Figure 3 at the end, where the authors have provided details about the graphs.  - Also for Figure's 2 and 4, some details about what can be seen in the graphs, not just their title

We have thankfully corrected the above and checked for other errors in the figure legends.

- In the Conclusions section: 

* I suggest that it needs to be improved because it does not present any conclusion of the present study, related to systemic free thiols.  * It is too general in the sentence: Increased systemic oxidative stress in eoFGR pregnancies is associated with the maternal syndrome...; there are many more biomarkers that can be investigated in the context of systemic oxidative stress.

We thank the reviewer for his/her comment. We acknowledge that FT levels is not exclusively a biomarker for systemic oxidative stress, for this current study we only were able to have FT as available biomarker. Therefore, we added in red.  ‘Increased systemic oxidative stress, as represented in this current study by FT levels in maternal serum samples, in eoFGR pregnancies is associated with the maternal syndrome, but not with the severity of fetal consequences of the placental dysfunction. These results help to understand the pathophysiology in eoFGR with and without maternal hypertensive disorders.’

Round 2

Reviewer 2 Report

Comments and Suggestions for Authors

The requested corrections have been made.